# Exploring Adverse Drug Reactions in Psychotropic Medications: A Retrospective Analysis of Portuguese Pharmacovigilance Data

**DOI:** 10.3390/healthcare12080808

**Published:** 2024-04-09

**Authors:** Ana Bandarra, César Costa, Kristina Angelova, Lília Leonardo, Margarida Espírito-Santo

**Affiliations:** 1Faculty of Sciences and Technology, University of Algarve, 8005-139 Faro, Portugal; a37145@ualg.pt; 2Unidade de Farmacovigilância do Algarve e Baixo Alentejo (UFALBA), University of Algarve, Campus de Gambelas, Building 2, Office no. 2.52, 8005-139 Faro, Portugal; cesar.costa@abcmedicalg.pt (C.C.); kristina.angelova@abcmedicalg.pt (K.A.); lilia.leonardo.ufaa@abcmedicalg.pt (L.L.); 3School of Health, University of Algarve, Campus de Gambelas, Building 1, 8005-139 Faro, Portugal; 4Algarve Biomedical Center Research Institute (ABC-RI), University of Algarve, Campus de Gambelas, Building 2, 8005-139 Faro, Portugal

**Keywords:** adverse drug reaction, pharmacovigilance, psychotropic drugs, spontaneous report

## Abstract

Psychotropic drugs (PDs) include anxiolytics, sedatives and hypnotics, antidepressants, and antipsychotics, and they are available as medicines with different safety profiles. Given Portugal’s high anxiolytic consumption and the rising prevalence of mental disorders, safety monitoring is crucial. This study aimed to analyze the individual case safety reports (ICSR) of suspected adverse drug reactions (ADRs) related to PDs, obtained through spontaneous reporting, and recorded in the Portuguese National Pharmacovigilance System between January 2017 and December 2021. This observational and retrospective study analyzed the ICSRs of suspected ADRs to PDs. Most reports pertained to female individuals (67.78%) between 18 and 64 years of age (63.71%). The pharmaceutical industry was the primary source of these reports (62.16%). Antidepressants were responsible for most ICSRs (61.90%). At least one serious ADR was recorded in 58.44% of the reports, and 43.84% of ADRs evolved into “cure”. The most-observed ADRs were nausea (10.92%), dizziness (10.70%), and off-label use (10.30%). In the causality assessment, 45.49% of ADRs were classified as “possible”, and only 4.96% were classified as “definitive”. The current analysis helps to strengthen the safety evidence for PDs. In the future, some measures could be implemented to improve the use of and/or access to PDs, as well as to reinforce the rate of suspected ADR reports within the community, contributing to the safety data available.

## 1. Introduction

Mental disorders (MDs) represent a significant public health concern, impacting approximately one in eight individuals globally. In Portugal, the prevalence of MDs is notably high, affecting one in five individuals. Among the most common MDs in Portugal are depression and anxiety disorders, with prevalence rates of 16.5% and 6.8%, respectively [1].

The prevalence of MDs in Portugal has shown a notable increase in recent years. According to the first National Epidemiological Health Survey conducted in 2013, the prevalence of MDs was recorded at 22.9% [2]. However, this figure has seen a rise to 27.6% according to the most recent study conducted in 2022 [3,4].

In a study conducted in 2021, 26.6% of the population aged 16 or over agreed that the COVID-19 pandemic brought a negative impact on the mental health of the Portuguese population [5].

Psychotropic drugs are a common treatment for MDs, such as anxiety disorders, depression, and psychosis. In Portugal, the consumption of psychotropic drugs has grown substantially in the last few years, with antidepressants and antipsychotics being the most prescribed. Between 2014 and 2022, the consumption of psychotropic drugs in Portugal increased from 12.9% to 15.9%, based on the National Health System expenditure per pharmacotherapeutic group, indicating a growing reliance on these drugs [6,7].

A wide range of side effects may arise from taking psychotropic drugs, depending on several factors, such as their mechanism of action, patient’s characteristics, and even the concomitant use of other drugs [8,9]. For example, the most common ADRs of antidepressants include nausea, vomiting, and headache. Frequently, the ADRs of anxiolytics include sedation, confusion, and drowsiness. Furthermore, ADRs to antipsychotics frequently encompass extrapyramidal symptoms and metabolic changes [10].

In general, ADRs related to the use of psychotropic drugs can be serious and may lead to hospitalization or, in very rare cases, to death. It is crucial to be mindful of the potential of the potential ADRs to these drugs, both before initiating treatment and throughout the course of therapy, as needed, to ensure patient safety [11].

In pursuit of this, it is crucial to conduct the ongoing surveillance of the safety of these drugs via passive or active pharmacovigilance methods.

In fact, pharmacovigilance is a critical aspect of drug development and post-marketing surveillance, ensuring the safety of medications for patients and encompassing a set of activities aimed at identifying, evaluating, and preventing adverse drug reactions (ADRs) [12,13].

In Portugal, the National Pharmacovigilance System (SNF) was established in 1992 and has undergone continuous evolution, adapting to the evolving landscape of drug development and usage. Initially, reporting was restricted to healthcare professionals; however, in 2012, the system expanded to all citizens. This system plays a pivotal role in monitoring ADRs, further enhancing the potential to capture a wider range of ADR experiences [14].

While there are studies on this topic at the international level, there is still a lack of research specifically focused on the Portuguese context.

Therefore, a study was performed with the aim of conducting a comprehensive analysis of individual case safety reports (ICSRs) of suspected ADRs associated with psychotropic drugs, including antidepressants, anxiolytics, sedatives, hypnotics, and antipsychotics.

## 2. Materials and Methods

This observational and retrospective study analyzed spontaneous reports of suspected adverse drug reactions (ADRs) to psychotropic drugs in Portugal, utilizing data from the Portuguese National Pharmacovigilance System (SNF) between January 2017 and December 2021.

Data were retrieved from INFARMED—National Authority of Medicines and Health Products, I.P.’s Portuguese electronic database of suspected adverse reactions, designated as Portal RAM.

The extracted data encompassed patient characteristics (age, sex, region according to the Nomenclature of Territorial Units for Statistics—NUTS II), reporter information (professional category), report details (year of reception, reporting method), drug-related information (international nonproprietary names—INN), anatomical therapeutic chemical (ATC) classification, pharmaceutical form, and ADR characterization (seriousness criteria; MedDRA system organ class (SOC) and lowest level term (LLT) classification; evolution; causality assessment). This study incorporated the following inclusion criteria: spontaneous reports received for psychotropic drugs (anxiolytics, sedatives, and hypnotics; antipsychotics; and antidepressants), received between 1 January 2017 and 31 December 2021, and involving individuals of both sexes and any age group. All cases originating from spontaneous reporting were considered, and reports from clinical studies were excluded. Duplicate, nullified, and rejected reports were also excluded. All drugs comprising the following classes of the Portuguese Pharmacotherapeutic Classification were included: 2.9—psychotropic drugs; 2.9.1—anxiolytics, sedatives, and hypnotics; 2.9.2—antipsychotics; 2.9.3—antidepressants.

Quantitative data were analyzed using descriptive statistics and presented as mean, median, standard deviation, minimum, and maximum. Qualitative variables were described as frequency (n) and percentages (%).

Prior to data collection, it was necessary to draft a study protocol, which was submitted to the Algarve Biomedical Center Health Ethics Committee (CES-ABC) and was approved on 23 June 2022.

Each ICSR analyzed involved at least one psychotropic drug that potentially was related to the ADR(s) reported. Nevertheless, there is still the possibility of more than one psychotropic drug being related to the suspected ADR(s).

## 3. Results

### 3.1. Characterization of Individual Case Safety Reports (ICSRs)

From January 2017 to December 2021, the Portuguese National Pharmacovigilance System (SNF) received a total of 1155 spontaneous reports of suspected ADRs including psychotropic drugs (anxiolytics, sedatives and hypnotics; antipsychotics; antidepressants), originating from various sources. These reports resulted in a total of 4197 ADRs, which involved a total of 1538 psychotropic drugs.

The age of patients included in the ICSRs under analysis varied widely, with a mean age of 53.22 ± 19.24 years. The youngest patient was 1 day old, while the oldest was 97 years old. No data were available for age in 15.06% (n = 174) of reports. Most reports were related to individuals aged between 18 and 64 years old (63.71%; n = 625), followed by individuals aged 65 or over (33.64%; n = 330) (Table 1).

Most of the ICSRs were related to female individuals (67.78%; n = 749), with no information available for this variable in 4.33% (n = 50) of cases. ICSRs were predominantly from the Lisbon Metropolitan Area (33.82%; n = 187), North (32.19%; n = 178), and Centre regions (20.98%; n = 116) (Table 1). It is important to note that, in the case of reports from the pharmaceutical industry (marketing authorization holder—MAH), the geographic location of these reports is blinded; hence, there exists a high number of reports for which the geographic origin is unknown.

Regarding the reporter, the ICSRs included in this analysis were carried out mostly (62.16%; n = 718) by the pharmaceutical industry. The contributions of health professionals such as pharmacists, physicians, and nurses were smaller (12.55%; n = 145) (Table 1).

Regarding professional specialties, the most frequent reporters were the community pharmacist (47.51%; n = 105) and the general practitioner (15.84%; n = 35), although this information was only available in 19.13% (n = 221) of the reports analyzed.

The highest number of ICSRs regarding psychotropic drugs occurred during the year of 2021 (25.37%; n = 293), and an increasing trend was observed throughout the period under analysis (2017–2021) (Figure 1).

Most of ICSRs related to psychotropic drugs were carried out by electronic transmission through Portal RAM (62.16%; n = 718). Considering those received at the Pharmacovigilance Regional Units (URF), the unit of “Lisbon, Setúbal, and Santarém” received the largest number of reports (14.03%; n = 162).

### 3.2. Characterization of Psychotropic Drugs Involved in the ICSRs

Considering the 1155 ICSRs analyzed in the current study, a total of 2814 medicines were identified, with an average of 2.44 ± 2.98 medicines used for each patient. Of these, more than half (54.66%; n = 1538) included psychotropic drugs (PDs), corresponding to 61 different PDs. At least one PD was identified for each report, and an average of 1.33 ± 0.85 PDs were identified per report, with a minimum of 1 and a maximum of 9 psychotropic drugs.

PDs identified in the ADRs were classified according to the WHO ATC Index [15] and the pharmaceutical form (PF).

The PDs more frequently included in the reports analyzed were antidepressants, followed by anxiolytics and antipsychotics (Figure 2).

The group of anxiolytics, sedatives, and hypnotics (N05B) displayed a notable prevalence in the reports, with drugs such as diazepam and alprazolam, both belonging to the class of benzodiazepines, accounting for 19.36% (n = 79) and 16.67% (n = 68), respectively, of the total number of drugs within this group (Table 2).

Antidepressant drugs, which were the most prevalent PDs in the reports analyzed in this study, featured sertraline and trazodone as the most commonly reported antidepressants, accounting for 18.59% (n = 177) and 16.81% (n = 160), respectively (Table 3). Antidepressive classes classified as “other antidepressants” and “selective serotonin reuptake inhibitors” were those more often involved in the reports under analysis.

Upon the evaluation of antipsychotic drugs, quetiapine (26.16%; n = 45) and olanzapine (15.12%; n = 26) emerged as the most frequently reported suspected drugs in the analyzed reports (Table 4).

The most common pharmaceutical forms identified in the analyzed reports were those intended for oral administration (97.43%; n = 606), with “film-coated tablets” and “tablets” being the most widely used, accounting for 36.82% (n = 229) and 23.15% (n = 144), respectively.

### 3.3. Characterization of Adverse Drug Reactions (ADRs)

All the reactions presented in the reports were analyzed for the variables used in the characterization of adverse drug reactions (ADRs), irrespective of whether there might be a relationship solely with the suspected psychotropic drugs or with other associated drugs in the same report.

For the 1155 ICSRs related to PDs, a total of 4197 ADRs were detected, indicating the presence of at least one ADR per report, with a maximum of 35 ADRs per report, and a median of 2 ADRs per report.

Most reports (58.44%; n = 675) contained at least one adverse drug reaction (ADR) classified as serious. 

Seriousness criteria were analyzed for ADRs classified as “serious”. Out of 675 reports (58.45%) indicating at least one serious ADR, a total of 987 ADRs were identified, with 45.63% (n = 308) classified as “clinically important” and only 4.30% (n = 29) categorized as “lethal”, resulting in death (Figure 3).

No cases of serious ADRs leading to “congenital anomaly” were observed.

Regarding the evolution of the total ADRs (4197 ADRs), about half (43.84%; n = 1840) evolved into a cure and only 2.12% (n = 89) resulted in death (Table 5). However, it should be noted that in each report, one or more different ADRs could be present, implying the existence in the same report of one or more ADRs leading to the same ADR outcome.

It is also important to highlight the number of ADRs for which the evolution was unknown, corresponding to 35.60% (n = 1494) of the reported ADR.

Using the classification of ADR by MedDRA system organ class (SOC) coding, an amount of 27 different SOC classifications were identified, with the most prevalent situations being “Nervous system disorders” (22.23%; n = 769) and “Psychiatric disorders” (16.53%; n = 572) (Table 6).

Out of the total 4197 ADRs related to psychotropic drugs identified in the current analysis, 1608 different “Lowest Level Term” (LLT) situations were identified, of which the most prevalent were “Nausea” (14.84%; n = 88), “Dizziness” (14.50%; n = 86) and “off-label use” (14.00%; n = 83) (Table 7).

Causality assessment, established by the regulatory authority using the World Health Organization Uppsala Monitoring Centre (WHO-UMC) causality system, was conducted for only 23.52% (n = 987) of the total identified ADRs. Of these, 45.49% (n = 449) were classified as “possible”, 42.25% (n = 417) as “probable”, and only 4.96% (n = 49) as “definitive” (Table 8).

Among the 49 ADRs classified as “definitive” by the regulatory entity, the majority involved antidepressive drugs (69.40%; n = 34), with the remaining cases encompassing anxiolytics, sedatives, and hypnotics (30.61%; n = 15) (Table 9).

Duloxetine and escitalopram were the psychotropic drugs that presented a greater number of ADRs classified as “definitive” based on the causality assessment.

## 4. Discussion

This study analyzed 1155 ICSRs of ADRs related to the use of psychotropic drugs, reported nationally between 2017 and 2021. These reports comprised a total of 4197 potential ADRs, encompassing 1538 psychotropic drugs.

In Portugal, within the National Health System (NHS), the consumption of medicines in 2021 revealed that drugs targeting the central nervous system (CNS) ranked as the second most consumed category, following medications affecting the cardiovascular system, which had the highest volume of consumption. Moreover, when examining NHS expenditure on medications in 2021, CNS-acting drugs ranked the second highest pharmacotherapeutic group by cost, accounting for 19.9% of the total expenditure within the NHS distribution of pharmacotherapeutic groups [7]. Also, the consumption of medicines containing psychotropic drugs showed an increasing trend throughout the period under analysis [16].

Considering the number of reports identified in the present study (n = 1155) in the period under analysis, reports related to psychotropic drugs account for only ~2% of the total reports received by the SNF during this timeframe [17].

These facts mentioned above would lead us to expect that the number of reports related to medicines belonging to these pharmacotherapeutic groups would represent a high fraction of the reports received by the SNF. However, this expectation was not met. In fact, the report number falls far below what was anticipated for these pharmacotherapeutic groups.

The current study found that individuals in the reports had an average age of 53.22 ± 19.24 years, with a greater number of reports within the 18 to 64 age group (63.84%). This aligns with data obtained from Portugal that analyzed ADR reports, indicating that adults and older adults are the groups with highest rate of adverse reaction reports [18,19].

Contrary to expectations based on the high consumption of medications, the group of older individuals did not exhibit a higher rate of reporting. Specifically, the group of older persons (65 years or older) presented a lower reporting rate compared to the adult group (33.64%). This could be attributed to the larger population size within the adult group compared to the older age group (≥65 years old) in Portugal. However, it is worth noting that the adult group may benefit from higher levels of health literacy compared to older age groups, which could influence reporting behaviors. Research indicates that health literacy tends to be lower among older individuals (≥65 years old) compared to younger age groups [20]. While age is not the sole determinant of health literacy, is the two factors are strongly correlated, with older age often associated with lower levels of health literacy [19,21].

The results showed a higher proportion of reports attributed to female individuals (67.78%), consistent with findings from other studies where the proportion of ADR reports from females ranged from 51.4% to 75.0% [18,19,22,23]. According to the literature, females tend to have higher consumption rates of PDs, which could contribute to the increased prevalence of ADRs observed in this study [24,25]. Additionally, according to data from the National Institute of Statistics (INE), Portugal had a population of 10,343,066 inhabitants in 2021, with females accounting for 52.43% of the total population [22].

Regarding geographical distribution, the majority of individuals experiencing ADRs hailed from the Lisbon (33.82%), North (32.19%), and Center regions (20.98%). These regions, according to the National Statistics Institute, have the highest numbers of inhabitants [22].

The pharmaceutical industry was the most significant contributor to reports, representing 62.16%, followed by pharmacists at 12.55%, and, subsequently, physicians with 12.21%. According to data provided by INFARMED—National Authority of Medicines and Health Products, I.P., from 2012 to the present, the pharmaceutical industry has consistently been recognized as a group of reporters showing a deep involvement in ADR reporting [26]. Considering the proximity of pharmacists, especially community pharmacists, to the general community, it would be expected that these healthcare professionals take on a more proactive role in reporting. However, factors such as insufficient information and time constraints have been identified as barriers hindering more active reporting of suspected ADRs [27].

Regarding physicians, who also made significant contributions to reporting potential ADRs, “general practice” emerged as the specialty with the highest level of involvement, possibly because it is the most accessible area for individuals within the community. The close relationship between physicians and patients during consultations could be a positive factor leading to the identification of suspected ADRs, thereby contributing to a higher rate of ADR reporting compared to other medical specialties [28,29].

According to INFARMED, I.P., physicians are the most common reporters of ADRs among healthcare professionals, followed by pharmacists [26]. However, in the current study, contrary to what is typically observed in the literature, pharmacists were the professional group that submitted a higher number of reports compared to physicians. This could be attributed to the proactive role of pharmacists and their proximity to the community. Additionally, physicians may face increased workloads and time constraints during consultations, which could impact their ability to report ADRs. [30].

Patients or other non-healthcare professionals were responsible for 11.86% of the reports, which is a low rate, considering data from 2014 onwards, which shows that these groups were responsible for ~30% of the total received ADRs [31]. This fact could be associated with a lack of awareness among the general population regarding the importance of reporting suspected ADRs and the potential benefits it holds for drug safety data [31,32].

The growing number of reporters, which now includes “patients or other non-healthcare professionals”, may be attributed to awareness campaigns conducted in the field of pharmacovigilance. These initiatives, spearheaded by INFARMED, I.P., in collaboration with URFs, have been developed in recent years and seem to be a program that will be further strengthened in the future, especially among groups that frequently use medication, such as older adults.

The evolution in the number of reports is in line with the rise in the consumption of PDs, according to a study conducted by the Portuguese Directorate-General of Health (DGH) between 2012 and 2016. This suggests a noticeable increase in the utilization of these drugs [33]. According to INFARMED, I.P., in a study monitoring the use of drugs in outpatient settings, psychotropic drugs were among the therapeutic classes with the highest consumption between January and March 2021. This includes anxiolytics, sedatives, and hypnotics (6.7%), antidepressants (6.3%), and antipsychotics (2.8%) [34].

According to the findings from studies conducted in Portugal, there has been a continuous and steady increase in the number of reports, which aligns with the observations in the current study between 2017 and 2021 [18,34]. During the peak of the pandemic in 2020, patients faced difficulties accessing healthcare appointments and facilities due to lockdown measures and resource reallocation toward COVID-19 [35]. Reduced healthcare-seeking behavior during 2020 could have affected the number of spontaneous ADR reports. Additionally, the global focus on COVID-19 during the pandemic may have led to decreased attention to other health issues, including the report of suspected ADRs, due to the prioritization of resources. It is noteworthy that there was a significant increase in the number of reports received during 2021, primarily related to COVID-19 vaccines [17].

Likely due to the increased reporting of suspected ADRs by non-health professionals in the last decade, there has been a steady improvement in the reporting rate. Nonetheless, despite this uptick, there remains a significant need to further bolster this intervention by all reporters.

In 2019, according to the National Health Council, Portugal ranked second among European countries concerning mental disorders, with a prevalence of 22.90%. This may help explain the high consumption of PDs in the country [36].

Regarding reporting suspected ADRs, this study revealed a significant number associated with medicines, including those belonging to the ATC group N (nervous system), where PDs constituted the third predominant group. These PDs accounted for 2/3 (13.63%) of the total medicines related to potential ADRs [23].

In the current study, the pharmacotherapeutic group with the highest prevalence of reports was antidepressants (61.90%), which is consistent with data from other countries in similar research areas [37]. It is estimated that depression affects 10% of the Portuguese population. Additionally, Portugal was ranked as the country in the world with the fifth highest consumption of antidepressant drugs [36,38,39]. Furthermore, considering the international landscape, the consumption of antidepressant drugs doubled in OECD countries between 2000 and 2017 [36].

Sertraline and trazodone emerged as the most frequently identified antidepressants in the analyzed reports, representing 11.51% and 10.40%, respectively. Between 2009 and 2018, the consumption of antidepressants in Portugal predominantly featured sertraline, trazodone, and venlafaxine as the most consumed drugs among the population, with a notable emphasis on sertraline [40]. This may explain the higher number of reports containing these drugs, as they are widely used compared to others in the same pharmacotherapeutic group. Another study conducted in Mexico pointed to sertraline as the antidepressant with the highest number of potential ADRs, representing 12.70% of ADRs [41].

In the pharmacotherapeutic group of anxiolytics, sedatives, and hypnotics, which accounted for 26.53% of the total psychotropic drugs related to the analyzed reports, a higher prevalence was observed for the drugs diazepam and alprazolam, both benzodiazepines, comprising 5.14% and 4.42%, respectively, of the total reports associated with PDs. Portugal is the European country with high consumption of psychotropic drugs, particularly anxiolytics such as benzodiazepines [42]. The increasing consumption of these drugs may result from treatments that extend beyond recommended durations and/or from their consumption for therapeutic indications for which they are not advised. This contributes to an increase in the number of potential ADRs in both scenarios [37].

According to report data from INFARMED, I.P., there was a noticeable upward trend in the consumption of PDs during 2021. This trend was characterized by a more pronounced increase in the use of antidepressant drugs, with a less marked rise in the consumption of anxiolytics, sedatives, and hypnotics [34].

Potential ADRs related to antipsychotics accounted for only 11.20% of the analyzed reports, representing a smaller proportion compared to other PDs. This may be attributed to their use in clinical situations with lower prevalence, such as bipolar disease and schizophrenia [3].

ADRs may arise from various factors, including drug interactions, individual characteristics, and medication errors. They represent a significant public health concern, with the potential to lead to unwanted negative outcomes [1,43]. In the current study, more than half (58.44%) of the suspected ADRs were considered serious, similar to data provided by INFARMED, I.P., on ADR analysis between 2012 and 2017 [18,44]. Similarly, studies conducted in Brazil analyzing ADRs identified through the country’s pharmacovigilance system found that approximately 59% of ADRs were classified as serious [45,46].

The most prevalent seriousness classification in this study was “clinically significant”, accounting for 45.63% of the reports identified as having resulted in at least one serious ADR. Another study conducted in Portugal, analyzing reports received between 2007 and 2017 and focusing on a different group of drugs (proton pump inhibitors), also showed a similar outcome with 58% of serious ADRs considered “clinically significant [44].

Approximately half (43.84%; n = 1840) of the flagged ADRs in the reports analyzed in the current study were resolved, while only 2.12% (n = 89) resulted in death. According to INFARMED, I.P.’s 2012 report, the analyzed ADRs were predominantly resolved, with values ranging between 78.5% and 80.5% [17].

The causality assessment for the ADRs identified showed that most were classified as “possible” (45.49%) and “probable” (42.25%), with only 4.96% classified as “definitive”. Another study conducted in Portugal, using data from reports received at the Porto regional unit, also observed that the classification of “definitive” was less frequently used for causality assessment (4.6%) [30]. The limited/unknown/incomplete information provided by reporters, as observed throughout this study´s analysis, may contribute to the low rate of ADRs classified as “definite”, as well as the difficulty in causality assessment, primarily due to the high number of drugs used. Additionally, the low rate of ADRs classified as “definitive” is closely related to the non-reintroduction of the drug, for ethical reasons, according to the WHO-UMC causality assessment tool.

The most prevalent ADRs identified, according to the MedDRA system organ class (SOC) classification, were “Nervous system disorders” (22.23%) and “Psychiatric disorders” (16.53%), aligning with the clinical area of the main pathologies treated with psychotropic drugs. Regarding classification by MedDRA lowest level term (LLT), the most prevalent were “Nausea” (10.92%), “Dizziness” (10.70%), and “off-label use” (10.30%). In reference to the first two, as per the literature, these are frequent ADRs across all classes of PDs, albeit they may not be classified as serious ADRs. As for “off-label use”, it can be defined as “the prescription of an authorized medication for a use not described in the Summary of Product Characteristics (SPC)”, and according to data available in the literature, PDs are increasingly used in this way in the field of psychiatry, especially antipsychotics, which have been widely used in the treatment of depression disorders [47,48]. Another example of off-label treatment is the use of antidepressant drugs in the treatment of sleep disorders, such as trazodone [49].

Considering the ADRs identified in the reports analyzed in the current study, the majority are already listed as possible ADRs of the medicines available on the market. However, even those that have been identified and are less frequent can still contribute to more comprehensive safety data on the use of PDs. Furthermore, in the future, enhancing the completeness rate of reports would be highly relevant, thus providing more evidence regarding potential ADRs and their relationship with the drugs commonly used by the population.

## 5. Conclusions

Between January 2017 and December 2021, Portugal received 1155 spontaneous reports of potential adverse drug reactions (ADRs) related to psychotropic drugs, including anxiolytics, sedatives, and hypnotics; antidepressants; and antipsychotics. These reports encompassed a total of 4197 ADRs associated with a total of 1538 psychotropic drugs. Over this period, there has been a noticeable increase in spontaneous ADR reports, indicating a positive progression of pharmacovigilance in Portugal.

Antidepressants emerged as the most frequent psychotropic drugs related to the reports analyzed (61.90%), with sertraline and trazodone as the primary contributors to these ADRs.

Nervous system disorders (22.23%) and psychiatric disorders (16.53%) were the most prevalent adverse drug reactions, according to the MedDRA SOC classification. Nausea (10.92%), dizziness (10.70%), and off-label use (10.30%) were the most frequent ADRs, using MedDRA LLT classification.

Given the widespread use of psychotropic drugs in Portugal, pharmacovigilance and the spontaneous reporting of ADRs are crucial areas contributing to public health. However, despite the extensive use of psychotropic drugs, particularly antidepressants and anxiolytics, sedatives, and hypnotics, there remains a lack of the spontaneous reporting of potential ADRs.

This analysis highlights the importance of fostering a more proactive approach to reporting suspected adverse reactions, involving healthcare professionals and the broader population. This approach aims to strengthen and build trust in the pharmacovigilance system, ultimately contributing to public health and medicines’ safety. This strategy is vital to consolidating the pharmacovigilance system and safeguarding public health.

## Figures and Tables

**Figure 1 healthcare-12-00808-f001:**
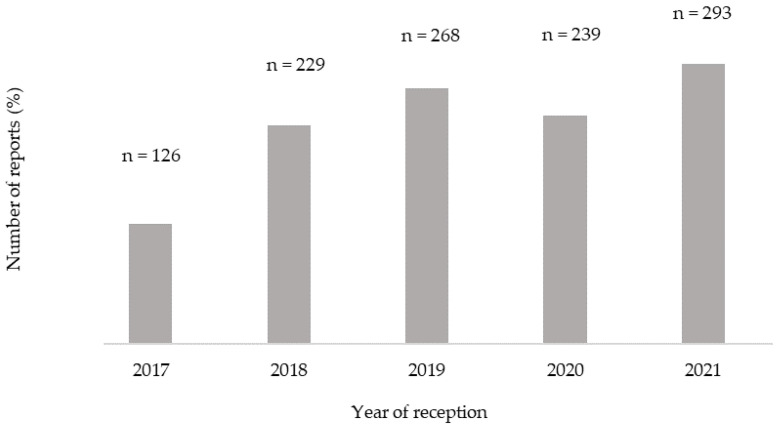
Characterization of the number of individual case spontaneous reports (ICSRs) related to psychotropic drugs between 2017 and 2021 in Portugal.

**Figure 2 healthcare-12-00808-f002:**
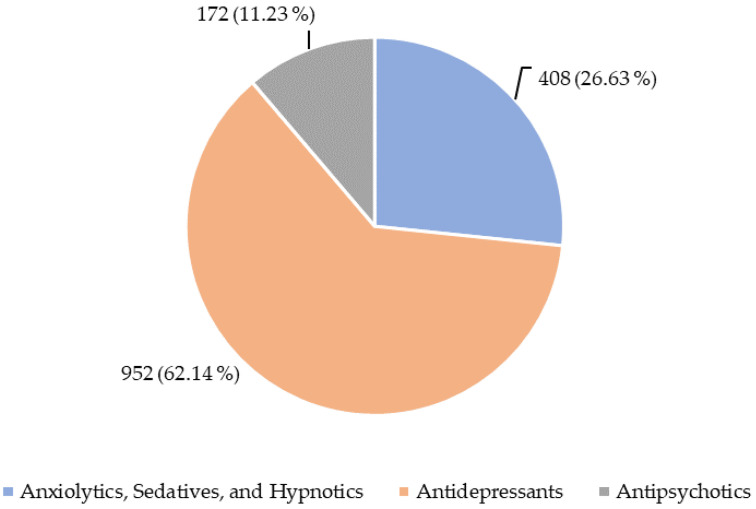
Classes of psychotropic drugs related to individual case safety reports (ICSRs) analyzed.

**Figure 3 healthcare-12-00808-f003:**
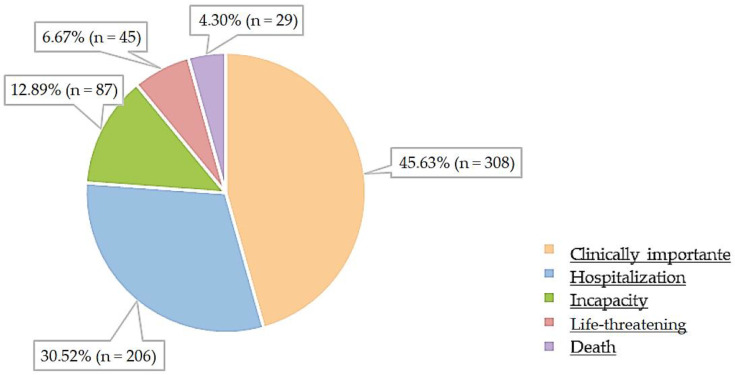
Classification of suspected ADRs according to seriousness criteria.

**Table 1 healthcare-12-00808-t001:** Characterization of patient’s age, geographic origin, and reporters of the individual case safety reports (ICSRs).

**Patient’s Age**	**% (n)**
0–1 month	0.31 (3)
2 months–2 years	0.10 (1)
3–11 years	0.31 (3)
12–17 years	1.94 (19)
18–64 years	63.71 (625)
More than 65 years	33.64 (330)
Unknown	15.06 (174)
Total	100.00 (981)
**Geographic Origin** (region according to NUTS II)	**% (n)**
North	32.19 (178)
Centre	20.98 (116)
Lisbon Metropolitan Area * West and Tagus Valley * Setúbal Peninsula *	33.82 (187)
Alentejo	5.79 (32)
Algarve	5.97 (33)
Autonomous Region of the Azores	0.72 (4)
Autonomous Region of Madeira	0.54 (3)
Unknown	52.12 (602)
Total	100.0 (1155)
**Reporters Professional Group**	**% (n)**
Pharmaceutical Industry	62.16 (718)
Pharmacist	12.55 (145)
Physician	12.21 (141)
Nurse	0.52 (6)
Other health professional (HP)	0.69 (8)
User or other non-HP	11.86 (137)
Total	100.00 (1155)

Comments: * These three regions were created in the most recent version of NUTS II (2024), and it is not possible to separate reports from previous dates according to this current geographic distribution.

**Table 2 healthcare-12-00808-t002:** Psychotropic drugs belonging to each ATC classification (3rd level).

Suspected Psychotropic Drug	ATC		% (n)
Alprazolam	N05B	Anxiolytics	16.67 (68)
Bromazepam			9.80 (40)
Clobazam			1.47 (6)
Chlordiazepoxide dipotassium			0.25 (1)
Cloxazolam			2.94 (12)
Diazepam			19.36 (79)
Ethyl loflazepate			4.66 (19)
Lorazepam			12.50 (51)
Oxazepam			2.45 (10)
Buspirone			0.98 (4)
Hydroxyzine			4.90 (20)
Total			75.98 (310)
Brotizolam	N05C	Hypnotics and Sedatives	0.25 (1)
Estazolam			0.49 (2)
Flurazepam			1.23 (5)
Midazolam			11.03 (45)
Temazepam			0.25 (1)
Triazolam			0.25 (1)
Valerian			0.98 (4)
Dexmedetomidine			1.72 (7)
Melatonin			0.25 (1)
Zolpidem			5.64 (23)
Total			22.06 (90)
Doxylamine	R06AA	Antihistaminic for systemic use	1.96 (8)
Total			1.96 (8)
Total			100.00 (408)

**Table 3 healthcare-12-00808-t003:** ATC classification (fifth level) for antidepressants related to the reports analyzed.

Suspected Psychotropic Drug	ATC Classification (5th Level)	ATC Classification (3rd Level)	% (n)
Imipramine	N06AA02	Non-selective monoamine reuptake inhibitors	0.11 (1)
Clomipramine	N06AA04		1.37 (13)
Amitriptyline	N06AA09		3.68 (35)
Nortriptyline	N06AA10		0.32 (3)
Total			5.48 (52)
Fluoxetine	N06AB03	Selective serotonin reuptake inhibitors	7.25 (69)
Citalopram	N06AB04		0.42 (4)
Paroxetine	N06AB05		4.10 (39)
Sertraline	N06AB06		18.59 (177)
Fluvoxamine	N06AB08		1.26 (12)
Escitalopram	N06AB10		9.87 (94)
Total			41.49 (395)
Moclobemide	N06AG02	Monoamine oxidase A inhibitors	0.11 (1)
Total			0.11 (1)
Pirlindole	N06AX	Other antidepressants	0.11 (1)
Mianserin	N06AX03		0.53 (5)
Trazodone	N06AX05		16.81 (160)
Mirtazapine	N06AX11		7.88 (75)
Bupropion	N06AX12		5.57 (53)
Tianeptine	N06AX14		0.53 (5)
Venlafaxine	N06AX16		9.03 (86)
Milnacipran	N06AX17		0.11 (1)
Duloxetine	N06AX21		5.36 (51)
Agomelatine	N06AX22		3.05 (29)
Vortioxetine	N06AX26		3.89 (37)
Esketamine	N06AX27		0.11 (1)
Total			52.08 (504)
Total			100.0 (952)

**Table 4 healthcare-12-00808-t004:** ATC classification (fifth level) for antipsychotics related to the reports analyzed.

Suspected Psychotropic Drug	ATC (Fifth Level)	ATC (Third Level)	% (n)
Cyamemazine	N05AA01	Phenothiazines with aliphatic sidechain	7.56 (13)
Chlorpromazine	N05AA01		2.33 (4)
Levomepromazine	N05AA02		3.49 (6)
Haloperidol	N05AD01	Butyrophenone derivatives	11.05 (19)
Zuclopenthixol	N05AF05	Thioxanthene derivative	0.58 (1)
Pimozide	N05AG02	Diphenylbutylpiperidine derivatives	1.16 (2)
Clozapine	N05AH02	Diazepines, oxazepines, thiazepines, and oxepines	4.07 (7)
Olanzapine	N05AH03		15.12 (26)
Quetiapine	N05AH04		26.16 (45)
Tiapride	N05AL03	Benzamides	1.74 (3)
Amisulpride	N05AL05		1.74 (3)
Risperidone	N05AX08	Other antipsychotics	13.95 (24)
Zotepine	N05AX11		1.16 (2)
Paliperidone	N05AX13		4.07 (7)
Aripiprazole	N05AX12		5.81 (10)
Total			100.0 (172)

**Table 5 healthcare-12-00808-t005:** Evolution of adverse drug reactions (ADR).

ADR Evolution	% (n)
Cure	43.84 (1840)
Unknown	35.60 (1494)
Recovery	8.98 (377)
Persists without recovery	8.53 (358)
Death	2.12 (89)
Cure with sequelae	0.93 (39)
Total	100.0 (4197)

**Table 6 healthcare-12-00808-t006:** Classification of the 10 most prevalent ADRs according to MedDRA SOC classification.

SOC Classification	% (n)
Nervous system disorders	22.23 (769)
Psychiatric disorders	16.53 (572)
General disorders and administration site conditions	14.45 (500)
Gastrointestinal disorders	14.42 (499)
Injury, poisoning, and procedural complications	9.16 (317)
Skin and subcutaneous tissue disorders	6.13 (212)
Metabolism and nutrition disorders	4.57 (158)
Musculoskeletal and connective tissue disorders	4.57 (158)
Investigations	4.51 (156)
Respiratory, thoracic, and mediastinal disorders	3.44 (119)
Total	100.00 (3460)

Abbreviations: ADR—adverse drug reaction; SOC—system organ class.

**Table 7 healthcare-12-00808-t007:** Classification of the 10 most prevalent ADRs according to MedDRA LLT classification.

LLT Classification	% (n)
Nausea	14.84 (88)
Dizziness	14.50 (86)
Off-label use	14.00 (83)
Headache	11.47 (68)
Vomiting	9.11 (54)
Diarrhea	8.60 (51)
Serotonin syndrome	7.08 (42)
Drug ineffective	6.91 (41)
Insomnia	6.91 (41)
Drug interaction	6.58 (39)
Total	100.00 (593)

Abbreviations: ADR—adverse drug reaction; LLT—lowest level term.

**Table 8 healthcare-12-00808-t008:** Assessment of the causality between the psychotropic drug and the adverse drug reaction (ADR).

Causality Assessment	% (n)
Definitive	4.96 (49)
Probable	42.25 (417)
Possible	45.49 (449)
Unlikely	4.56 (45)
Conditional	0.51 (5)
Unclassifiable	1.32 (13)
Not related	0.71 (7)
Unknown	0.10 (1)
Not applicable	0.10 (1)
Total	100.00 (987)

**Table 9 healthcare-12-00808-t009:** Psychotropic drugs having causality assessment classified as “definitive”.

Psychotropic Drug	ADR According to MedDRA LLT Classification	% (n)
Duloxetine	Tightness of jaw muscles; vomiting; swelling of tongue; nausea; blood pressure increased; malaise; numbness of limbs	20.41 (10)
Escitalopram	Tremor; tachycardia; agitation; anxiety; confusion; insomnia; hypotension; intentional overdose; somnolence; altered state of consciousness	20.41 (10)
Sertraline	Chapped lips; tremor; sexual dysfunction; diarrhea; dry mouth; malaise; raw gums	14.29 (7)
Lorazepam	Prostration; drowsiness; hypotension; intentional overdose; somnolence; altered state of consciousness	12.24 (6)
Diazepam	Hallucination; hypotension; incoherent; bradycardia	8.16 (4)
Hydroxyzine	Angioedema; hallucination; maculo-papular rash; altered state of consciousness	8.16 (4)
Fluoxetine	Anorgasmia; libido decreased	4.08 (2)
Agomelatine	Pimple-like rash	2.04 (1)
Alprazolam	Panic attack	2.04 (1)
Mirtazapine	Agitation	2.04 (1)
Paroxetine	Urticaria localized	2.04 (1)
Trazodone	Urinary tract obstruction	2.04 (1)
Venlafaxine	Medication in stool	2.04 (1)
Total		100.00 (49)

## Data Availability

Data are contained within the article.

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
