# Peer review of "Exploring Adverse Drug Reactions in Psychotropic Medications: A Retrospective Analysis of Portuguese Pharmacovigilance Data"

_healthcare, 2024, doi:10.3390/healthcare12080808_

Round 1

Reviewer 1 Report

Comments and Suggestions for Authors

The manuscript entitled, "Exploring Adverse Drug Reactions in Psychotropic Medications: A Retrospective Analysis of Portuguese Pharmacovigilance Data" is a critical survey of the adverse effects of psychotropic drugs pertaining to the Portuguese Pharmacovigilance Data.

1.      In the introduction section, the lines 28-20 are very generic, the very first and second lines. Authors should cite the relevant literature to support the relevant data. 2.      Authors should make a paragraph of 5-7 sentences rather than 1-2 sentences throughout the manuscript.

3.      In Table 1, authors should correct the data presentation, 0 a 1 month to 0-1 month.

4.      Table 2 may be presented in the pie chart format as a Figure for better presentation.

5.      Table data is interesting. However, authors should make the comments why the most drugs have been taken by 60 plus patients.

6.      In figure 1, authors should discuss why the pattern is not followed for the year 2020?

7.      References are not written as per the author's guidelines. Some references are not even in the English language. Authors should adhere to the standard guidelines.

Author Response

Dear Reviewer,

Thank you for your work, as we believe that all your suggestions will improve the quality of the article that we are now resubmitting for publication.

Please find below our point-by-point responses to all the suggestions and comments. Changes to the manuscript are shown under track changes.

Best regards,

Reviewer 1:

 The manuscript entitled, "Exploring Adverse Drug Reactions in Psychotropic Medications: A Retrospective Analysis of Portuguese Pharmacovigilance Data" is a critical survey of the adverse effects of psychotropic drugs pertaining to the Portuguese Pharmacovigilance Data.

Thank you so much for your comment and recommendation for publication.

  1. In the introduction section, the lines 28-20 are very generic, the very first and second lines. Authors should cite the relevant literature to support the relevant data.

Thank you for the suggestion, the changes have been carried out.

  1. Authors should make a paragraph of 5-7 sentences rather than 1-2 sentences throughout the manuscript.

 The text was revised in general, and an effort was made to follow this suggestion.

  1. In Table 1, authors should correct the data presentation, 0 a 1 month to 0-1 month.

Thank you for the suggestion, the changes have been carried out.

  1. Table 2 may be presented in the pie chart format as a Figure for better presentation.

Thank you for the suggestion, indeed, it now have a better presentation.

  1. Table data is interesting. However, authors should make the comments why the most drugs have been taken by 60 plus patients.

 Unfortunately, we were unable to identify which table you are referring to. Later, we will be available to perform the suggested changes.

  1. In figure 1, authors should discuss why the pattern is not followed for the year 2020?

 The explanation for this situation was included in lines 378-384.

  1. References are not written as per the author's guidelines. Some references are not even in the English language. Authors should adhere to the standard guidelines.

 All references were reviewed and presented in accordance with the journal's guidelines.

Reviewer 2 Report

Comments and Suggestions for Authors

I find the work very good and recommend its publication. There are a few issues with the English language and I appreciate that English is not the first language of the authors. I must accept that it is not my first language either. I find one or two instances in which I find the need for changes, I am not sure whether the authors really meant what is written or is it because of the language usage that I have misunderstood it. I believe a thorough editing of the language will rid the paper of most inconsistencies.

  Comments on the Quality of English Language

There are quite a few instances of faulty construction of sentences, though an informed reader can make sense of what the authors  want to convey,  sometimes it is a challenge. For example;

The mean age of patients involved in the SRs under analysis was 53.22 ± 19.24 years old, with a minimum of 1 day and a maximum of 97 years.

Here the word "involved" is not the right one and need to be replaced, the word "old" is redundant, and needs to be deleted.

Hence there is need for editing with respect to English language. Unfortunately, since the manuscript is presented in a pdf form, editing will be an onerous task for me.

In some cases, the words fail to convey the proper meaning, for example:

In 2014, the consumption of psychotropic drugs was 12.9%, and an increase to 15.9% in 2022 was observed [6,7].

I fail to understand what this figure represents, 12.9% of what?

The authors conclude that older people (above 65 years) are not associated with the maximum number of SRs. I contest this on the basis that the population of people over 65 is lower than that of people between 18 and 64 (see Table 1), hence comparing these two groups is misleading.

Please see the order of Causality in Table 9. I suggest the causality to be listed in order of increasing or decreasing association.

Author Response

Dear Reviewer,

Thank you for your work, as we believe that all your suggestions will improve the quality of the article that we are now resubmitting for publication.

Please find below our point-by-point responses to all the suggestions and comments. Changes to the manuscript are shown under track changes.

Best regards,

Reviewer 2:

I find the work very good and recommend its publication. There are a few issues with the English language and I appreciate that English is not the first language of the authors. I must accept that it is not my first language either. I find one or two instances in which I find the need for changes, I am not sure whether the authors really meant what is written or is it because of the language usage that I have misunderstood it. I believe a thorough editing of the language will rid the paper of most inconsistencies.

Thank you so much for your comment and recommendation for publication.

Comments on the Quality of English Language

There are quite a few instances of faulty construction of sentences, though an informed reader can make sense of what the authors  want to convey,  sometimes it is a challenge. For example;

-The mean age of patients involved in the SRs under analysis was 53.22 ± 19.24 years old, with a minimum of 1 day and a maximum of 97 years.

In fact, data included in this sentence may seems strange. Nevertheless, the age of patients included in the reports under analysis varied widely, with a mean age of 53.22 ± 19.24 years. The youngest patient was 1 day old, while the oldest was 97 years old. This information was reformulated (lines 114-118).

A general review of the English language within the manuscript was carried out.

-Here the word "involved" is not the right one and need to be replaced, the word "old" is redundant, and needs to be deleted.

Thank you for the suggestion. The word “involved” was replaced by “included” and “old” has been removed.

-Hence there is need for editing with respect to English language. Unfortunately, since the manuscript is presented in a pdf form, editing will be an onerous task for me.

In some cases, the words fail to convey the proper meaning, for example:

In 2014, the consumption of psychotropic drugs was 12.9%, and an increase to 15.9% in 2022 was observed [6,7].

I fail to understand what this figure represents, 12.9% of what?

The sentence was reformulated: “Between 2014 and 2022, the consumption of psychotropic drugs in Portugal increased from 12.9% to 15.9%, considering the expenditure of National Health System per pharmacotherapeutic group, indicating a growing reliance on these drugs” (lines 41-46).

-The authors conclude that older people (above 65 years) are not associated with the maximum number of SRs. I contest this on the basis that the population of people over 65 is lower than that of people between 18 and 64 (see Table 1), hence comparing these two groups is misleading.

The results of the current study just realize that the number of reports from older persons is lower than that of individuals aged between 18 and 64 years. In fact, the number of individuals is lower for this age group (over 65 years), despite the higher number of medications used by them, which represents a greater risk for the occurrence of ADRs. This discussion´s point was reformulated (line 312 – 317).

- Please see the order of Causality in Table 9. I suggest the causality to be listed in order of increasing or decreasing association.

Table 8 (former Table 9) was reformulated, and results were presented in order decreasing of causality association.

Reviewer 3 Report

Comments and Suggestions for Authors

Dear Colleagues

It was a pleasure to read your manuscript. It is always interesting to read a pharmacovigilance study.

Please consider the following notes:

-            Although the article per se is interesting and relevant, the approach is not new. Please detail more carefully the importance of the study;

-            The article is written by authors that work/studies in an university that trains both Pharmacy technicians and Pharmacists. Pharmacy technicians are not specifically mentioned in the context of reporting or being involved in the spontaneous reports of adverse drug reactions related to psychotropic medications. Please review this.

-            Please check your circular graphic, the subtitle says “importante” isntaed of “important”

-            Legendas should be captions or subtitles and not “legends”

-            What is the framework/catalogue/rule you used when describing the severity of the adverse drug reactions?

-            Did you consider polypharmacy situations and implications in the reports?

-            Are there any data on the evolution of the prescription of these drugs that can be used to enhance your results?

Author Response

Dear Reviewer:

Thank you for your work, as we believe that all your suggestions will improve the quality of the article that we are now resubmitting for publication.

Please find below our point-by-point responses to all the suggestions and comments. Changes to the manuscript are shown under track changes.

Best regards,

Reviewer 3:

Dear Colleagues

It was a pleasure to read your manuscript. It is always interesting to read a pharmacovigilance study.

Thank you so much for your comment and recommendation for publication.

Please consider the following notes:

-  Although the article per se is interesting and relevant, the approach is not new. Please detail more carefully the importance of the study;

Thank you for the suggestion, the changes have been carried out within the introduction chapter (lines 31-72).

 - The article is written by authors that work/studies in an university that trains both Pharmacy technicians and Pharmacists. Pharmacy technicians are not specifically mentioned in the context of reporting or being involved in the spontaneous reports of adverse drug reactions related to psychotropic medications. Please review this.

The online platform (Portal RAM), available in Portugal to report suspected ADRs, in the reporter field, only allows to select one of the following categories "Doctor, Pharmacist, Nurse, Other Health Professional, User or Other Non-Health Professional”. Therefore, unfortunately we are unable to quantify the reports provided specifically from other health professionals besides those already reported.

  - Please check your circular graphic, the subtitle says “importante” instead of “important”.

Thank you for the suggestion, the changes have been carried out.

 - Legendas should be captions or subtitles and not “legends”.

Thank you for the suggestion, the changes have been carried out (e.g. Table 1, Table 5, Table 7).

 - What is the framework/catalogue/rule you used when describing the severity of the adverse drug reactions? 

The seriousness of ADRs was classified according to WHO criteria. A serious ADR can be classified according to 5 different criteria, which are in increasing order of severity: clinically important, when they disrupt the patient's normal daily routine; cause congenital anomaly/malformation; results in temporary or permanent disability; requires or prolong hospitalization; life-threatening; or cause death.

 - Did you consider polypharmacy situations and implications in the reports?

We are aware that polypharmacy is a risk factor for the occurrence of adverse reactions, but in the current work we did not aim to analyze this specific association. In the future we would like to explore this situation with more detail, exploring drugs most involved in ADRs and respective mechanisms involved.

 - Are there any data on the evolution of the prescription of these drugs that can be used to enhance your results?

Prescription data are not available, but statistical data on consumption show an increasing consumption trend over the period under analysis. This information was included in lines 368-374.